# Designing a Proximal Sensing Camera Acquisition System for Vineyard Applications: Results and Feedback on 8 Years of Experiments

**DOI:** 10.3390/s23020847

**Published:** 2023-01-11

**Authors:** Florian Rançon, Barna Keresztes, Aymeric Deshayes, Malo Tardif, Florent Abdelghafour, Gael Fontaine, Jean-Pierre Da Costa, Christian Germain

**Affiliations:** 1IMS Laboratory, CNRS UMR 5218, University of Bordeaux, Talence Campus, F-33400 Talence, France; 2Bordeaux Sciences Agro, F-33175 Gradignan, France; 3INRAE, Institut Agro, ITAP, University of Montpellier, F-34196 Montpellier, France

**Keywords:** precision viticulture, smart farming, proximal sensing, disease detection, yield estimation, image analysis, mobile camera, deep learning

## Abstract

The potential of image proximal sensing for agricultural applications has been a prolific scientific subject in the recent literature. Its main appeal lies in the sensing of precise information about plant status, which is either harder or impossible to extract from lower-resolution downward-looking image sensors such as satellite or drone imagery. Yet, many theoretical and practical problems arise when dealing with proximal sensing, especially on perennial crops such as vineyards. Indeed, vineyards exhibit challenging physical obstacles and many degrees of variability in their layout. In this paper, we present the design of a mobile camera suited to vineyards and harsh experimental conditions, as well as the results and assessments of 8 years’ worth of studies using that camera. These projects ranged from in-field yield estimation (berry counting) to disease detection, providing new insights on typical viticulture problems that could also be generalized to orchard crops. Different recommendations are then provided using small case studies, such as the difficulties related to framing plots with different structures or the mounting of the sensor on a moving vehicle. While results stress the obvious importance and strong benefits of a thorough experimental design, they also indicate some inescapable pitfalls, illustrating the need for more robust image analysis algorithms and better databases. We believe sharing that experience with the scientific community can only benefit the future development of these innovative approaches.

## 1. Introduction

Extensive insight about the properties of a field has always been a key asset for farmers. The appropriate management of a plot is conditioned by the historical knowledge regarding yields, sanitary status, frost episodes or soil properties such as texture or organic matter content. Crop protection, fertilization and other cropping calendar operations can then benefit greatly from the precise knowledge of spatial and temporal patterns in the plots. Human abilities are however limited, preventing the exhaustive knowledge on each plant in the field and the adequate use of that knowledge. The sensing of the environment to gain new knowledge, whether it is qualitative or quantitative (leading to the precision agriculture paradigms [1]), has thus become a new challenge in modern agriculture.

As an answer to those challenges, new sensing technologies have been progressively deployed in agriculture, using either invasive or noninvasive methods [2]. In particular, we are interested here in the subject of optical sensing. Optical sensing allows the capture of an image of the plant, which can be useful to estimate one or several key parameters of interest such as phenological properties, structure vigor or yield. Optical sensing uses a nonspecialized type of sensor, adaptable to many different applications. The only condition is that the phenomenon of interest must be captured by the image. It has thus gained a lot of traction these last decades thanks to recent technological progress (picture resolution, image quality, sensor size).

It is however important to keep in mind that many types of cameras exist, with different purposes and underlying agricultural applications. RGB cameras are commonly used for their high spatial resolution at a relatively low price. They are suited for tasks where a difference in hue, shape or texture is enough to extract relevant information. Considering the simplicity of this approach, an extensive literature is available on the subject, examples ranging from disease classification [3] to automatic delineation of cadastral maps [4]. Other types of more complex and expensive sensors have been used, namely multispectral [5,6] and hyperspectral [7] cameras, stacking more than three spectral bands in the visible domain (ranging from four to hundreds of bands) in order to characterize plant health. Studies using multispectral or hyperspectal cameras tend to heavily use the red-edge region or the water absorption regions of the near-infrared spectrum [8]. In that case, correcting the obtained spectra into standardized reflectance spectra is a significant challenge, meaning these approaches are less suited for field experiments with natural uncontrolled lighting.

On another note, the vector on which the camera is mounted is as important as the sensor itself. Vectors are often separated on the basis of the distance between the sensor and the plant, often referring to satellite- and UAV-based acquisitions as remote sensing and ground-based acquisition as proximal sensing. Depending on the flight height, drones may be seen as an intermediary and thus may be labeled as both remote and proximal sensing. Satellite imagery has been extensively used for its high spatial footprint, allowing for a complete examination of the plots at various scales. The recent launch of Sentinel-2 satellites made this approach even more popular, thanks to its high temporal resolution [9] and the free access to its data. The low spatial resolution however implies it is unsuited for many applications, because many objects of interest cannot be properly distinguished when the resolution is too low. Cloud coverage also means perfect temporal continuity is not possible.

Proximal sensing can be subdivided into two types of vectors: mobile and static. The former allows one to cover a plot at a single date [10] while the latter allows one to build temporal series of a single point in the plot [11]. Some acquisition systems also make use of LIDAR sensors [12] or RGB-D cameras’ [13] abilities to map the surrounding environment, raising other methodological issues about the fusion of imagery and 3D point clouds. For instance, the information provided by depth cameras can be used to create more accurate yield predictions [14]. Given the wide array of existing sensors, the choice of the right architecture for optimal proximal sensing then becomes extremely important. While remote sensing provides the opportunity to observe plants from above, proximal sensing is about the observation from the side at different angles. This is relevant for perennial crops grown with a foliar hedge architecture. Indeed, the majority of the information (fruits, disease symptoms) often resides in the trellising plan.

For more information on the various uses of optical sensing in agriculture, the reader may refer to the review by [15] on the subject of remote sensing. To our knowledge, however, no comprehensive review has been created on the subject of optical proximal sensing in viticulture.

Optical proximal sensing can be seen as a two-step process: the first one is the acquisition (including the whole experimental design) and the second one is the image processing to convert the image into the desired results. The processing step may be synchronous to the acquisition or may be done later. The former is obviously more challenging but is key to time-critical operations such as weeding [2]. Image processing as a whole is also seen as a challenging part, combining scientific fields such as signal processing, machine learning and deep learning [16] in order to create specific and complex algorithms suited for a task (e.g., detecting the number of fruits [17]) or a set of tasks (differentiating the symptoms from different diseases [3]). However, the acquisition step is also known to be challenging since it takes place in noncontrolled environments. Natural lighting can only be partly controlled and thus has a definitive effect on the obtained pictures, which may or may not be possible to correct at a later stage. In addition, field hazards such as dust, heat and physical obstacles, also make things more difficult (risk of damaging the sensor, lack of mobility of the vehicle in the field, weather conditions, sensor overheating, etc.). Sometimes, unexpected and nondetectable issues may even make a dataset unusable (faulty GNSS system, camera lens obscured by dust).

However, it can be difficult to get into the details of these problems, considering the wide range of optical sensors, crop types and vectors that can be used. The agricultural and scientific community actors have thus specialized on very specific research domains, in order to gain field experience and design the best-suited acquisition systems. Scientific validation and promotion of these works is typically done through research articles covering the methodology and the results, most of the time with a description of the acquisition system. Data publishing is also more and more encouraged, leading to more occurrences of articles in which the original data are freely made available for the community to use [18]. More details about the availability of the individual datasets presented in this study can be found in the Data Availability Statement section.

The choices made during the acquisition system design step tend to be less discussed in the literature, even though they are critical to the success of the application. A discussion about the choice of the most appropriate vehicle for proximal sensing in agriculture can be found in [19]. Still, that study remains general and does not focus on the specificity of the grapevine crop. Depending on the context and the application, other grapevine studies may discuss matters related to vehicle choice, lighting and image framing. In many cases, artificial light is used to control the scene lighting, using, for instance, LED units [20,21]. In [21], experiments using the lighting unit during night and day were also conducted. The authors concluded night image acquisitions were better to enable standardized conditions, especially when the color of objects of interest (such as berries) was important. Flash illumination methods have also been reported in the literature [22]. In that berry counting application, two flash units with a custom diffusion filter were mounted on both sides of the camera, the distance between the camera and the fruiting zone was about 0.9 m and 1.5 m. Other authors chose not to rely on artificial lighting, meaning a correction may have been necessary to account for natural variability [10]. In that study, the authors proposed a radiometric calibration method in order to standardize the multispectral images used to compute in-field vegetation indices.

A direct use of agricultural machinery is also reported in some studies [23]. There, a field phenotyping platform (aiming at yield and vigor parameters estimation) was directly integrated to a grapevine harvester, taking advantage of the tunnel structure to avoid disturbing environmental factors such as sunlight effects. Additionally, that acquisition setup used a broadband light source to enforce near-standardized lighting conditions. One other originality of that work was the use of three vertically aligned cameras, allowing them to cover the canopy of each vine (the study also reported the distance from the camera to the canopy was approximately 0.75 m). More compact and original vehicles have also been reported in the literature, such as a within-row caterpillar equipped with an RGB-D sensor [24] or a quad [20]. In both studies, only the grape area was targeted.

While many studies focus on the vine plant and thus use a lateral camera mounting point, others are also interested in weed mapping or general vineyard navigation. In that case, the sensor may be mounted on the front side of the vehicle, facing the direction of the vehicle movement. A compact robotic platform dedicated to weed mapping using this setup and an RGB-D sensor was for instance reported in [13]. Other studies chose not to use a vehicle and conducted data acquisition using a handheld phone or a camera mounted on a pole [25]. In that case, the framing and image quality are easier to control, but the data acquisition speed is a concern.

Finally, the incidence of experimental design choices on the efficiency of recent deep learning algorithms was also discussed in a review comparing different plant disease detection studies ranging from laboratory experiments to field experiments [3]. In that research work, data diversity and contextual features in images were seen as the key parameters to achieve robust disease detection results.

The case studies and projects presented in this paper come from both a research lab specialized in image and signal processing perspective (IMS “Intégration du Matériau au Système” laboratory—UMR 5218, CNRS Talence, France) and from an agronomic engineering school perspective (Bordeaux Sciences Agro, Gradignan, France). Our past experiments led us to believe mobile proximal sensing was a promising solution for agricultural applications needing high-resolution imagery and a full view of the plant, such as fruit counting or disease detection. Our localization (south-west of France, Bordeaux region) also naturally led us to work on the grapevine crop, which is the main focus in this paper. Other works on vegetable weeding applications or orchard trees [26] are not mentioned, even though some lessons may still hold true in the case of orchard trees. Moreover, only color (RGB) imaging is discussed. A single RGB camera was used for all the undermentioned projects; full details about the sensor can be found in the Materials section. A wide variety of vehicle types has been tested over the years and is also discussed in this paper. Finally, while experiments outside the grapevine vegetative season are interesting for pruning applications, we chose to discard them from the study.

The main objective of this paper is thus to discuss, based on our previous experiments on various applications, the challenges and opportunities related to the use of vehicles equipped with RGB cameras in the vineyard. We believe the things we learned and the hardness we overcame may be of interest for the proximal sensing scientific community.

In chronological order, four projects are used in this paper. Project EARN’s main objective was to estimate yield in a vineyard. As a first step, automatic berry counting was performed and compared to manual counting [27]. Later projects aimed at detecting disease symptoms on vines, namely grapevine wood diseases (project ADVANTAGE [28]), downy mildew (project ProxiMaVi [29]) and flavescence dorée (project ProspectFD [30]). These three disease detection projects differed by their end goals. For instance, flavescence dorée is a damaging epidemic disease, meaning sources of contamination must be quickly detected. In that case, proximal sensing is a promising way to quickly detect and contain outbreaks (uprooting). Downy mildew detection is mostly related to spraying management, which is a costly operation with high environmental impact. Finally, esca is a major cause of long-term wilting in grapevine with no curative treatment, and early detection of symptoms may allow viticulturists to quickly replace diseased plants and limit losses. Table 1 presents a synthetic view of the four projects with complementary references. To our knowledge, no other work on in-field downy mildew detection using optical proximal sensing exists.

**Table 1 sensors-23-00847-t001:** Description of the scientific projects used in the study. When possible, references of past scientific publications are provided. (* publication covering both grapevine and apple detection applications).

Project Name	Application and Image Analysis Task	Years	Project Literature Reference	Other Literature References on the Application (In-Field Optical Proximal Sensing)
EARN	Yield estimation, berry counting	2014–2016	[27] *	[31,32]
Prospect FD	Flavescence dorée, epidemic management (uprooting), detection of symptoms	2020-present	[30]	[25,33]
ProxiMaVi	Downy mildew management (spraying), detection of symptoms	2016–2019	[29]	-
ADVANTAGE	Esca disease management, detection of symptoms	2015–2018	[28]	[33,34]

The paper is organized as follows:

In the first part, the acquisition system used for all the previously mentioned projects is presented, both from a hardware and a software perspective. The integration and mounting of the system on different vehicles are discussed, as well as the organization and the processing of the obtained datasets. It is important to keep in mind individual results and performance metrics of the different developed image processing algorithms are not discussed in detail, as the reader may refer to the individual published papers of each project for more information.

In the second part, the incidence of experimental design parameters on the resulting datasets and applications is discussed using small case studies.

Finally, recommendations and insights about how future work may handle these issues are provided.

## 2. Materials and Methods

### 2.1. Hardware Architecture

The main inspiration behind the sensor was first and foremost the numerous practical constraints when performing in-field image acquisition. This means the ability to control, as much as technically possible, the acquisition environment. The second inspiration was the range of potential applications. Given this variability, the sensor’s architecture had to be versatile with interchangeable and optional features so it could be easily adapted to the various use cases. Easy maintenance and emergency repairability were also important.

The sensor architecture (as shown in Figure 1) was controlled by a single board computer. Depending on the application, Raspberry Pi model 4 or Nvidia Xavier/Nano computers may be used. The more powerful Nvidia computer was only used when edge computing was necessary and is not discussed here. An independent battery-operated clock was connected directly to the Raspberry Pi, allowing it to keep track of the date and time (as there is no clock built into the Raspberry Pi). The 64 GB on-board storage was enough to store around 50,000 images. For comparison, taking one picture per vine would result in around 10,000 pictures per hectare in a vineyard with narrow 1.1 m width rows. In that case, this meant roughly 5 ha could be covered in a single acquisition.

The sensor was equipped with a 5-megapixel RGB industrial camera, the Basler Ace acA2440-20gc, which features a global shutter (well-suited for capturing moving objects) 2/3${}^{\prime \prime }$ type CMOS sensor and a Gigabit Ethernet connection. We used 6 mm or 8 mm optics with a 85${}^{\circ }$ and 69${}^{\circ }$ diagonal field of view, respectively. At the typical acquisition distance of 1 m, the physical resolution of a picture was about 4 pixels/mm^2^. It is important to note that the typical distance may not be achieved in practical conditions; results shown later in this paper illustrate this phenomenon.

An important feature of the sensor is the controlled illumination. A powerful xenon Phoxene SX3 flash provided enough luminosity to counter natural light (output power up to 20 J). This allowed a constant luminosity across the images, without shadows. A powerful lighting also led to a brief exposure time, eliminating the motion blur that may be caused by the vibrations of a moving vehicle; a smaller aperture, resulting in a deeper depth of field; and a “day for night” effect that illuminates the foreground and keeps the background dark. Practical issues arise when repeatedly using a flash (for instance, once every second) during prolonged periods of time. For this purpose, an industrial xenon flash was chosen for our sensor architecture. Additionally, one major advantage of this setup was the ability to synchronize the flash and the camera, which enabled a brief exposure time (approximately 250 µs).

The sensor used a GNSS system to georeference the images and control the acquisition process. Depending on the required precision, we could use the ublox Zed-F9P high-precision unit or a standard precision generic USB GNSS unit based on the ublox Neo-6 chip. Since real-time georeferencing was not necessary, postprocessed kinematics (PPK) could be used to achieve subdecimetric precision, which was sufficient to geolocalize vines. The system could be powered by its own 12 V batteries or connected to 3 pin DIN 9680 sockets. The DIN 9680 socket applies to tractors and agricultural machinery with connector systems for transmitting electrical power, up to 25 V. This meant the system could be easily mounted on and connected to agricultural tractors.

If needed, an optional ultrasonic rangefinder (Maxbotix HR-USB-EZ1) could be added to the sensor to measure the distance from the vines and detect the missing ones. The distance data were used to determine the area covered by the photo. These data could also be combined in real time with the GNSS positions for a better control of the acquisition process. For instance, acquisition could be stopped when too many missing plants were detected by the sensor or when the system left the acquisition zone, reducing power consumption, the number of irrelevant images and preserving the flash.

As a whole, one key advantage of the system was the ability to easily plug different devices (camera, flash unit, GNSS system, rangefinder) on a standardized central unit (Raspberry Pi 4+). Connections were also kept minimal, namely a USB port (GNSS module), a serial port (flash unit) and an Ethernet connection (camera).

In addition, the system could be embedded in different housings; two examples are showcased in Figure 2. Housing changes allowed us to adapt to different agricultural contexts (for instance, using a more compact case for narrow rows to avoid any contact with vegetation or poles, as shown in Figure 2b).

The sensor was designed to simplify the integration with different vehicles; some examples are shown in Figure 3. As the agricultural machines do not feature standardized mounting points for sensors, specific mounting systems needed to be created for every vehicle, whether it was traditional agricultural machinery (Figure 3b–d) or custom robots (Figure 3a). The use of steel channels could also be necessary in order to adjust the camera field. Given the variability of situations and machinery in the agricultural world, we believe this is an unavoidable issue when working with different vineyards. It is also worth noting the sensor could not be mounted on a vehicle and could be used as a pedestrian system (using a custom cross-shaped steel beam support), allowing quick experiments in the vineyard.

That consideration for flexibility also meant the sensor was suited to different vineyards and crop managements, or even other crops such as apple trees. This way, it was possible to use the same architecture with few adaptations for different projects. For example, it was possible to quickly change the camera sensor or add another type of sensor (such as the rangefinder) to complete specific needs for the data acquisition.

### 2.2. Software Architecture

The sensor was controlled by the embedded single-board computer. A piece of software handled the camera, the user interface, the GNSS module, the storage device and the image processing pipeline. As the embedded Raspberry Pi computer was not particularly powerful, the piece of software had to be highly optimized, mainly by parallelizing the different processes to run in parallel on separate CPU cores.

As the sensor can be hard to reach during the image acquisition, the main operation mode was totally autonomous. In this case, the image acquisition was started and stopped using the positioning data. The active plots could be defined in any geographical information systems (GISs), including Google Earth, as polygons and saved as “.kml” files. As the sensor entered one of these plots, the acquisition process started and stayed active as long as the sensor was inside the plot.

The sensor could also be controlled manually using a remote Wi-Fi connection. An Android smartphone application was developed (Figure 4), which connected to the sensor. The user interface was exhaustive, showing the status of every component (Figure 4a), the live captured image (Figure 4b), free storage space and other information. It also allowed us to start and stop the acquisition manually and to set the camera parameters, such as exposure time and frequency, as well as managing the data (deleting, copying to or extracting on a USB device, updating the software tool, etc.).

The embedded piece of software analyzed the acquired images for over- and underexposure and other errors. If needed, it could also process the images in real time using deep neural networks and other GPU-based computer vision algorithms on the Nvidia embedded computers.

### 2.3. Data Storage

The image files were stored in jpeg format (85% compression, around 1 megabyte). Each image was linked to a text file containing geolocalization, timestamp and distance information (when the rangefinder was plugged in). The files were temporarily stored on the sensor then transferred to a server equipped with a high-end GPU. Datasets were organized using a simple PostgreSQL spatial database, keeping track of the different pictures (combined with the GNSS position when possible) and their associated plots/sensors/vehicles.

### 2.4. Image Processing Summary

The projects presented in this study were aimed at different applications and used different processing methods. Berry detection (project EARN [27]) used as a first step a radial Hough transform to highlight spherical shapes. The illumination power of the embedded flash was used to create light gradients toward the center of each berry, which could then be detected. As a second step, a convolutional neural network (CNN) was used to invalidate potential false positives. Each berry could then be identified by its center and its size. Disease detection projects used both conventional machine learning approaches and more recent deep learning approaches. Image foreground extraction followed by texture analysis with a color structure tensor was used to detect downy mildew spots [29], while detection neural networks were used to detect the position of esca and flavescence dorée in the pictures using bounding boxes [28,30]. In the case of flavescence dorée detection, a random forest algorithm was also trained on the leaves detection to predict whether or not the whole vine suffered from flavescence dorée.

A summary of that information, as well as visual examples of end results, can be found in Table 2. More details about these projects, the databases and the quantitative evaluation of the results can be individually found in the associated published papers (Table 1).

While the individual examination and evaluation of the used algorithms is beyond the scope of this paper, it is worth noting both traditional handmade features extracting algorithms and state-of-the-art deep learning algorithms were used, and in some cases compared [28]. While traditional methods tend to lag behind in terms of raw performance, we believe handmade features may be more robust in some cases, and their potential shortcomings with different datasets may be easier to predict. Even though each algorithm used the original image of 2448 pixels × 2048 pixels as the basis, downscaling was always performed as a subsequent step (in order to alleviate GPU memory issues with deep learning frameworks). Details about the labeling process of each project can also be found in the associated papers.

All these experiments were conducted separately without any overlap. There is a growing interest in performing the simultaneous detection of several grapevine symptoms on the leaves (or on the grapes). However, this leads to two major problems.

The first one is obviously methodological. Designing an algorithm suited to two different problems can be a huge challenge, even when using more flexible deep learning architectures. For instance, downy mildew and esca disease lead to foliar symptoms at different scales. Mildew leads to small spots and esca leads to global patterns on the leaf. This also means the end result can be different, as illustrated here since esca was detected using bounding boxes and mildew spots were precisely segmented.

The second one is purely practical. Different diseases cause symptoms on the plant at different time periods, which do not necessarily overlap. In our case, it was not possible to detect both mildew and esca at the same time, the latter being expressed at a much later time during the grapevine vegetative season. Detecting both esca and flavescence dorée is, however, possible [30]. Simultaneous disease detection and yield estimation would lead to similar problems.

However, the simultaneous detection of the plant architecture and of symptoms is an interesting prospect. In the case of flavescence dorée, the lack of wood hardening and/or grapes adds precious contextual information and may be crucial to the decision [30].

The following section is dedicated to the description and the discussion of three critical parameters when conducting experiments in vineyards. The application-specific differences when dealing with these parameters are also discussed.

## 3. Results

After examining the datasets and the results for the four projects in the study, the acquisition parameters with an effect on the obtained images (and a potential effect on the end result) were roughly separated into three categories:Acquisition system integrity;Parameters influencing the appropriate framing of the object of interest;Parameters related to lighting and colors.

Problems related to GNSS data were omitted from that list. One major issue with geolocalization is that, even when access to subdecimetric RTK/PPK precision is possible, the camera position does not amount to the vine position. A geometric transformation is possible but the distance between the camera and the plant (as well as the view angle) may not be constant during the acquisition. This means converting a position dataset to a lattice vine position grid can be tricky. Even though they are critical for some applications, GNSS logs are auxiliary data that do not directly contribute to the integrity and the quality of the obtained pictures. Hence, we chose not to get into the details of these.

In the following sections, the three categories are described and analyzed, using the experience from the projects as a basis.

### 3.1. Acquisition System Integrity

Integrity parameters are the most obvious when considering the whole experimental chain. They tend to be critical since they can prevent the picture data recording. As explained in the description of our acquisition system, they are mostly related to the environment (e.g., dust, humidity or heat) and randomly hinder experiments. Adequate protection of the acquisition system is key to avoid these (e.g., reinforced case and connectors). It is however important to keep in mind it is nearly impossible to entirely avoid the effect of the environment. The effect of the acquisition vehicle is also crucial. For instance, an acquisition system mounted at the bottom of a harvesting high-clearance machine would get less affected by direct sunlight than if it was mounted on the side of a tractor. However, it would also get more affected by dust. Severed cables and wrong voltages may also randomly happen and threaten individual parts (e.g., flash device) or the whole acquisition system. Spare parts and basic repairing tools thus have to be brought accordingly into the field during the experiments.

Integrity parameters effects tend not to be application specific. The only exception to this would be a faulty flash device for applications relying on direct light reflections (e.g., berry counting); however, all tested applications greatly benefited from a working flash and the resulting improved contrast in pictures.

### 3.2. Framing-Related Parameters

When using proximal sensing, it is essential for the area of interest to be part of the camera frame. In fruit crops organized as rows, however, this may be a challenge, especially in vineyards with very narrow rows. Looking at Figure 5, it is easy to see the framing quality may drastically vary between acquisitions in the vineyard. Lower and higher parts of the vertical trellis may be missing from the picture. Missing the lower part (Figure 5c) is obviously a problem for applications such as yield estimation. Not dealing efficiently with framing issues makes yield estimation a more difficult task since that bias needs to be corrected, on top of existing biases such as berries occluded by leaves.

In some cases, the side parts of the plant may also be missing. Depending on the crop, the pruning system and the within-row distance between plants, the borders between neighbor plants are also sometimes hard to distinguish as illustrated in Figure 5c, where two different vines blend together. This factor should be monitored vigilantly for applications where precise per-plant diagnosis is necessary (e.g., individual uprooting operations for grapevines affected by chronic esca symptoms). Other ones such as yield estimation or global epidemic diagnosis (flavescence dorée) suffer less from overlapping plants in the pictures. Possible partial solutions to horizontal framing problems may include plant-by-plant manual acquisition or a precise use of RTK GNSS positions combined with previous knowledge of the vines’ positions.

In order to ensure the pictures are well framed, one could use the following simple formula to compute the appropriate distance for a given height: (1)$$d=\frac{f*h}{s},$$
where *f* is the focal length (mm), *s* is the height of the sensor (mm) and *h* is the physical height of the object (mm). Using this formula, it is easy to compute the recommended distances to achieve a proper framing in a vineyard, as illustrated in Figure 6. For instance, if *d* = 50 cm, a surface represented by the orange dotted square can be covered. In that case, most of the foliar area can be sampled. However, this does not guarantee perfect framing, since any vertical shift from this perfect configuration will make it subpar.

Looking at the possibilities, it is obviously impossible to always control the framing during the acquisition, as it would imply that the camera could be placed at any distance of the plant, which is not true for row crops such as vineyards, in which the environment can sometimes be very tight. When using a tractor or a movable robot, the vehicle is running in the middle of the row with the camera facing one row, meaning the maximum possible distance is inferior to the width of the row. For instance, a lot of vineyard rows in France are separated by roughly a single meter. Using a high-leg tractor may alleviate the issue but in that case, the maximum distance still is inferior to the distance between rows. Another possible trick is moving the camera to the front or the back of the vehicle in order to move the camera away from the plant as much as possible, with no risk of machinery parts appearing in the frame (setup illustrated in Figure 3a).

Even when the distance between rows is known, it is crucial to also consider that part of that space will be used by the plant foliage volume and clearance within high-vigor rows may be more difficult. When taking into account this volume and the spatial footprint of the acquisition vector, the maximum distance quickly becomes limited. It is then often mandatory to make compromises in the framing. Obviously, the estimations made in Figure 6 only hold true for this specific example, as parameters such as the distance between plants and the height of the plant at a specific phenological stage for a specific pruning system vary a lot. A lot of these parameters are actually linked, vigor being for example a complex combination of cultivar effect, pruning system, local soil and plant health. Unfortunately, it is not trivial to take this into account, vigor being a phenomenon with huge spatial and temporal variations [35]. This means predicting the optimal framing setup beforehand is almost impossible, and most of these settings need to be done on the fly before the experiment.

One possible alternative to accommodate for narrow rows may be changing the optics; fisheye cameras could for instance be considered in more extreme cases. However, this implies using different cameras, adding variability in the image database and specific image treatments (distortion correction). When working with several projects, this is not a desirable behavior since algorithms will have to be designed accordingly. It is also worth noting border effects may occur, related to the casing or the camera optics. Trained algorithms may thus also fail to operate in these areas.

Another important consequence of the framing variability is the objects’ size variability in the resulting images. Even though image size is constant, objects appear bigger when the camera is close to the plant. That, however, does not necessarily mean they are easier to detect for the trained algorithms. If we add to this variability the natural seasonal variability (phenological stages BBCH71 through BBCH89 in the case of grapevine berries [36]) and cultivar differences, huge differences are to be expected. As a small visual experiment, Figure 7 presents a set of sample square patches of leaves (Figure 7a) and berries (Figure 7b) extracted from all the study projects.

Again, the importance of algorithm robustness shines here. Even in the same geographical area and at the same date, it is not wise to expect objects with similar sizes. A multiplicative coefficient up to three should be expected in object sizes when sampling other vineyards. This means the algorithm need to be able to perform under degraded conditions, which may be impossible for some applications (e.g., small downy mildew spots).

Appropriate vertical framing can also be difficult to achieve when dealing with agricultural machines since the mounting point may be too low or too high. In theory, choosing a vehicle depends on the vineyard characteristics and on the surface we need to survey. In reality, the availability of the vehicles and conductors is actually the most important part, as summarized in Table 3. Tractors need a tractorist and harvesting machines are only available for specific operations, meaning they are only suited for certain applications at a short period of time. On the other hand, small robots and do-it-yourself solutions can be used at any time but have a more limited work rate.

In a nutshell, this means that unless the experiment plot is very well known and a tailored solution can be imagined to perfectly fit it, perfect framing is near impossible for large-scale experiments. This also implies research efforts devising algorithms robust to these problems and estimating the bias of models fitted with that kind of uncertainty (on top of already-present uncertainty, e.g., grapes occluded by leaves in very dense vines) are a top priority in proximal sensing.

Another consequence of noncontrolled environments is the possible inclusion of abnormal objects in the field of the camera, or normal objects obstructing the field. Occlusion is bound to happen, especially when the camera is mounted on the lower part of the acquisition vector (e.g., harvesting machine). Weeds (Figure 8b) or even sometimes full leaves (Figure 8a) may obstruct the object of interest. Processing of the picture then becomes more difficult, or even impossible in some cases.

Depending on the algorithm and the training base, abnormal objects may also trigger false positives. An example of this phenomenon is shown in Figure 9. In that case, the objective was to differentiate esca symptoms from other grapevine symptoms and deficiencies (for instance, dried leaves). Here, an abnormal brown object filled with weeds was wrongly detected as two dried leaves. This was not a surprising error since no such objects were present in the training set. It is wise to expect many other potential false detections of this kind are possible. An algorithm may for instance detect pheromone dispensers as dried leaves, even though it did not happen in our studies.

These false positives tend to be rare and may or may not compromise the underlying application. In the case of yield estimation, exceptional false berry detections have no significant incidence when compared to the total number of berries and the expected yield precision. Disease detection may be a bit more problematic, but the problem can be avoided by considering, for instance, the total number of detected symptomatic leaves in each image [30], only detecting a disease on the plant when the number of detected leaves is superior to a given threshold.

### 3.3. Illumination Related Parameters

In a similar way to the parameters influencing framing, lighting parameters have an influence on the picture quality and are unavoidable in noncontrolled environments. As explained in the sensor presentation part, the use of a flash device allows us to mitigate the effects of natural lighting. However. it has to be fine-tuned for the current lighting conditions, which is a challenge in many practical cases. On cloudy days, illumination may randomly switch between direct and diffuse lighting. Light levels also rapidly change between dawn and morning, which can be a problem when doing experiments on several plots over the course of a few hours. Figure 10 presents an example of a time series of pictures taken throughout the morning on the same plot. Darkening of the background gradually becomes more and more difficult and the general color palette gradually changes.

Bad flash timing or nonfunctional flash are also causes that may influence the illumination of the scene. Yield estimation can be particularly affected by these issues as it relies on the detection of spherical illuminated objects. Figure 11a presents an example of nondetected berries caused by a badly adjusted flash while Figure 11b presents an example of nondetected berries caused by a faulty flash. Flash usage sometimes also causes a small light halo on the borders of the picture.

The consequences when failing to darken the background using the “day for night” technique are variable and depend on the application. In the case of small berries detection, the consequences are usually small, since background objects appear very small and out of focus. This can, however, be a problem for disease detection, especially flavescence dorée and esca detection (leaf-scale symptoms). Figure 12 presents two samples from the esca detection dataset. In Figure 12a, background leaves are not detected by the algorithm, while in Figure 12b, one background leaf is wrongly detected as esca.

## 4. Discussion

In the previous part, we described the various parameters during acquisition with an effect on the resulting images and a potential effect on algorithm efficiency. As a summary, Figure 13 arranges that list of parameters according to two axes: whether or not the parameter is problematic and whether or not it can be controlled by the user.

This figure shows unavoidable problems usually are not the most problematic. However, unavoidable problems (related to environmental effects) are still bound to happen randomly. Application-specific problems ask for cautious planning, even though part of them cannot be fully mastered, especially those related to the environment.

If we consider the acquisition system presented in this paper and the ones presented in other research works, it is quickly apparent direct comparisons between studies can be challenging. While similar design decisions can be encountered (such as the use of a flash unit to detect spherical fruits [22] or the use of high-leg tractors [23]), many design choices are actually related to vineyard layout and management. For instance, among the cited research works, two of them provided information about the distance between vine rows [22,23]. In both cases, that distance was superior to 75 cm, meaning the row width was less of an issue than in our studies. Similarly, other researchers may encounter specific problems that prevent them from using certain vehicles such as heavy agricultural machinery. This is a possible issue in grapevine plots with steep slopes or with wet soil conditions. Another example of this diversity is vine pruning. Depending on pruning and thinning practices, grapes may grow at a constant height [20] or spread on the vertical plan [21]. These differences significantly affect image framing and experimental design.

It is thus expected the proximal sensing scientific community and commercial market will face significant challenges in the future when trying to adapt existing computer vision algorithms to different vineyards. While proximal sensing has attracted a lot of attention these last 10 years, great precautions are needed when analyzing the results and imagining future commercial applications. Indeed, whether they try to roughly detect or precisely segment objects and phenomena on plants, algorithms will always have to deal with a huge variability and unexpected cases. It is safe to assume the vineyards sampled in this paper do not accurately describe the actual range of vineyard variability: worse performance is then expected when applying the algorithms to other vineyards. That being said, it is however important to keep in mind the observations made in this paper are related to the diversity of French vineyards, which may be less important in other vine-growing countries. Its great diversity of cultivars and pruning systems makes it harder to design acquisition systems suited for all the vineyards, even in the same vine-growing region (Bordeaux).

The acquisition issue is closely related to the database size and variability problem. The most robust algorithm cannot be trained without an extensive database showcasing detailed spatial (from the region scale to the plot scale) and temporal (phenological stages) variability. Ways to automatize picture acquisition are thus crucial, notably, finding ways to systematically link agricultural machine passage (tractors, high-leg tractors, robots, etc.) with picture acquisitions. This, however, means algorithms must be designed around huge image databases which, in the current research state, tend to favor deep-learning-based algorithms. In this paper, the projects’ database sizes varied a lot (depending on the vehicle and the number of available vineyards) but the actual numbers of labeled images used to train the algorithms was similar. The image labeling step is thus a clear bottleneck here. This means existing pipelines to label the training images need to be strengthened to be more efficient, whether they use fully manual annotation or new techniques such as transfer learning (TL) or few-shot learning (FSL) [37]. The specificity of manual annotations in the agricultural world is, however, that in certain cases, the user needs to be highly skilled at the task at hand. Actually, even skilled professionals may struggle when labeling disease images, due to the lack of contextual information. For instance, flavescence dorée labeling benefits a lot from grapes and shoots examination. Whole pictures of the plants at different angles would be optimal in that case, which stresses even more the importance of parameters related to picture framing.

While the focus of this paper was the vine crop, similar conclusions could be derived when dealing with orchard crops such as apple trees. However, several key differences have to be taken into account, such as wider rows, higher objects of interest and different agricultural machinery uses. Similarly, experimenting on grapevines without leaves during the winter season implies different strategies, since background removal becomes crucial when considering potential wood-pruning applications. Using complementary depth information from sensors such as RGB-D cameras is a promising way to ensure the background is efficiently removed [38].

Future works may include the combination of different applications in order to obtain a more contextual diagnosis of the plant. They may also include the combination of mobile proximal sensing with fixed sensors, allowing us to merge high-spatial-resolution data with high-temporal-resolution data. The wealth of spatial information generated by the experiments (e.g., disease detection maps) may also lead to the creation of spatial models, allowing us to gain a better understanding of the studied phenomena (epidemic diseases, vigor variability, etc.).

## 5. Conclusions

In this paper based on 8 years of field experiments, we described the importance of experimental parameters when conducting proximal sensing acquisitions using a movable vector in vineyards. Organ detection and disease detection examples from four projects were used to evaluate which parameters are crucial to the success of potential applications, as well as which parameters can be controlled by an adequate experimental protocol.

Results showed that even though part of the acquisition workflow can be controlled, great variability and unexpected behaviors will always be expected, such as erratic framing, varying light conditions or random material failures. This means any processing algorithm should be, from the get-go, designed to take into account these differences. Recent approaches thus tend to favor more robust deep learning frameworks. However, given the sheer complexity of vineyards and well-known “black box” effects related to deep learning, we believe significant efforts in the understanding and the evaluation of these algorithms’ robustness will be necessary in the future to properly scale up proximal sensing applications.

Results also stressed the need for the creation of better databases representing the huge variability of configurations encountered in vineyards, a phenomenon with further repercussions on the choice of algorithms. Mounting the cameras on agricultural machinery in order to get repeated images from each plant in the vineyard is a promising approach in order to boost the augmentation of these databases. However, it highlights a lot of practical issues, related to the camera mounting point, farmer cooperation or the use of the device in extreme conditions (spraying, moving mechanical parts). These issues still hold true in the case of autonomous robots, whose increasing uses may be partly a solution for the automation of image-collecting tasks.

## Figures and Tables

**Figure 1 sensors-23-00847-f001:**
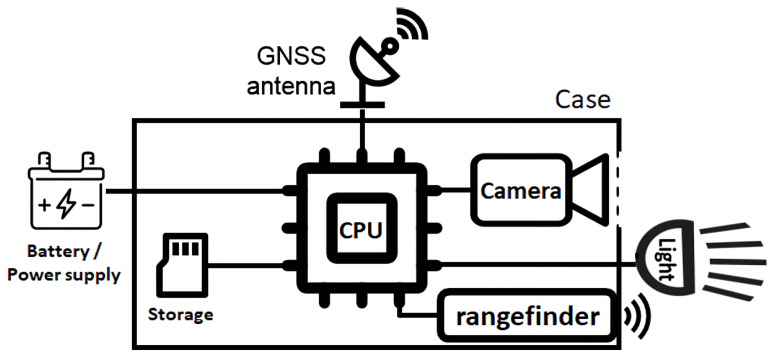
Simplified schematic representation of the acquisition system.

**Figure 2 sensors-23-00847-f002:**
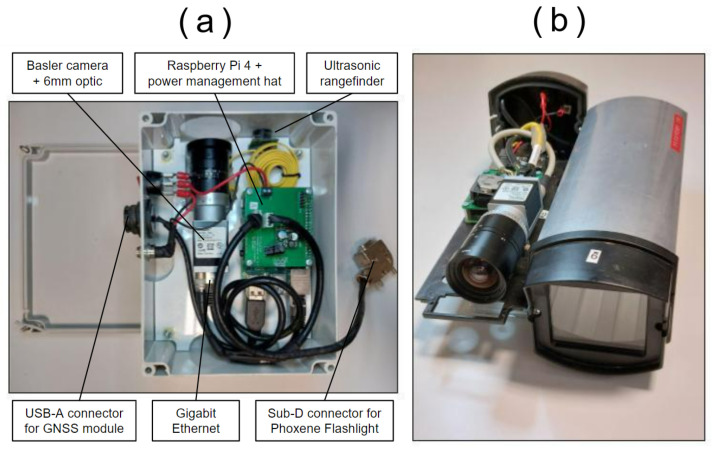
Photography of the sensor embedded in two different cases. (**a**) Large case (including details about devices and connections); (**b**) Narrow case.

**Figure 3 sensors-23-00847-f003:**
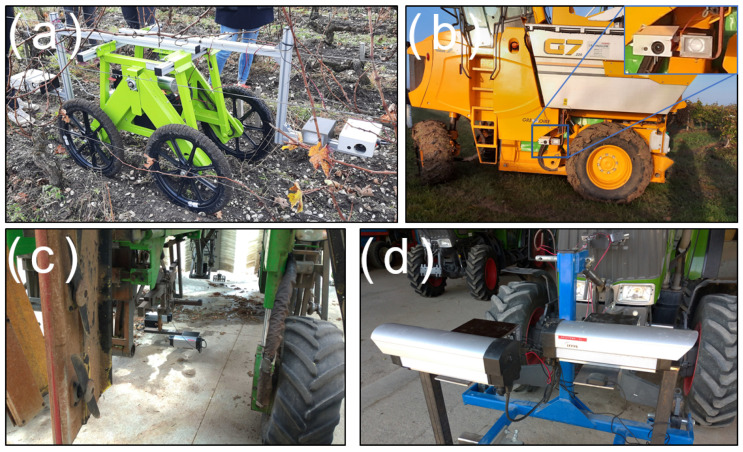
Photos of the sensor mounted on different vehicles. (**a**) Mounted on a small electric robot; (**b**) mounted on a harvesting machine; (**c**) mounted under a straddle tractor; (**d**) mounted on a tractor.

**Figure 4 sensors-23-00847-f004:**
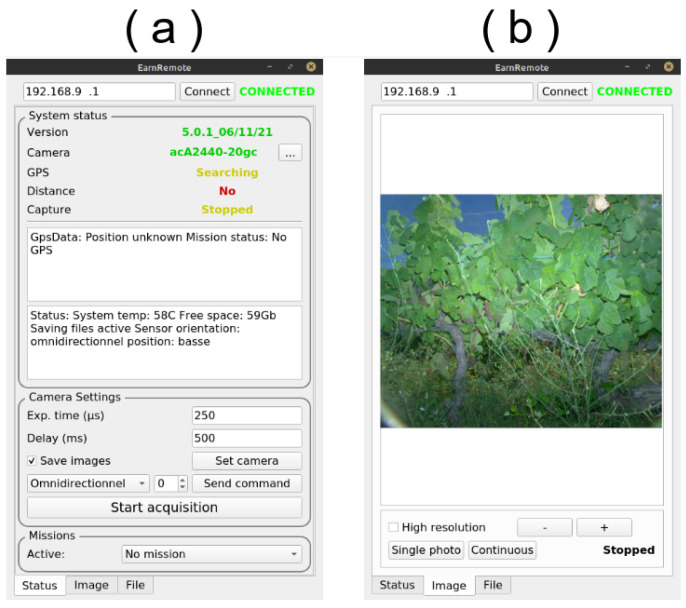
Screenshots of the in-house Android smartphone application used to control and monitor the sensor. (**a**) Connection and settings. (**b**) Picture quality preview. Overexposed or underexposed parts of the picture are highlighted in red when present.

**Figure 5 sensors-23-00847-f005:**
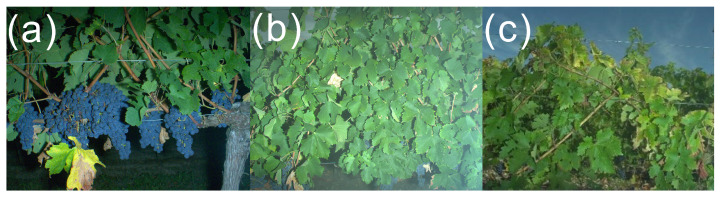
Examples of framing differences on grapevine acquisitions. (**a**) Bottom part and trunk of the vine. (**b**) Leaf area of the vine. (**c**) Top part of the vine. Images extracted from the 4 projects’ datasets.

**Figure 6 sensors-23-00847-f006:**
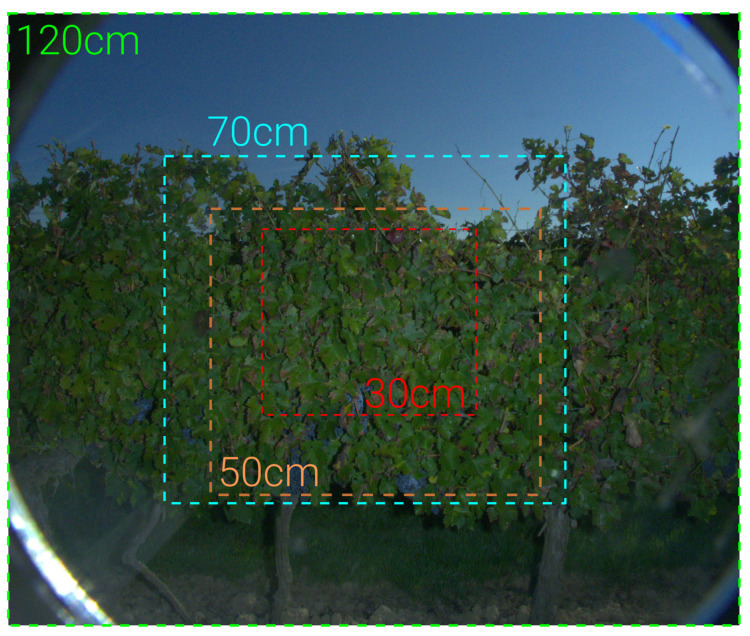
Examples of framing possibilities on one vine row. For each framing example (color dotted rectangles), the corresponding distance between the camera and the vertical trellis is indicated with the same color.

**Figure 7 sensors-23-00847-f007:**
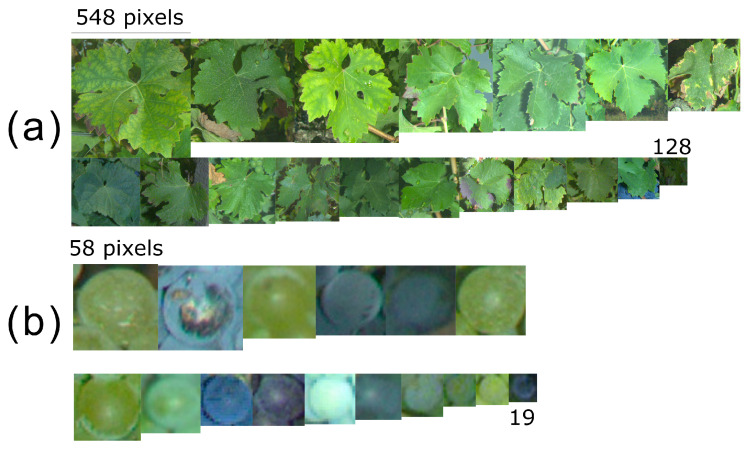
Samples of leaf (**a**) and berry patches (**b**) in the databases from the 4 study projects. Numbers indicate for each category the minimum and maximum square patch size in the batch. Original full images were all of size 2448 × 2048 pixels.

**Figure 8 sensors-23-00847-f008:**
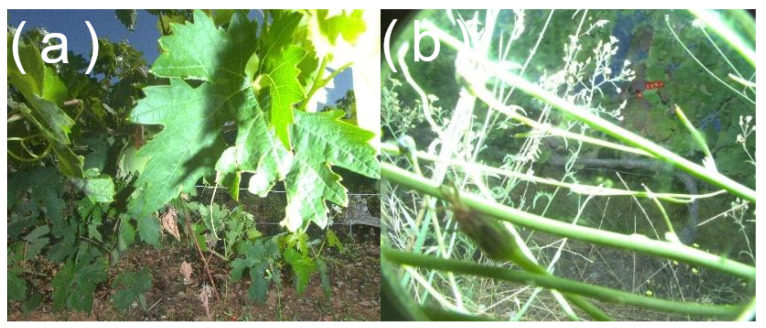
Example of images showcasing occlusion by (**a**) leaves and (**b**) weeds.

**Figure 9 sensors-23-00847-f009:**
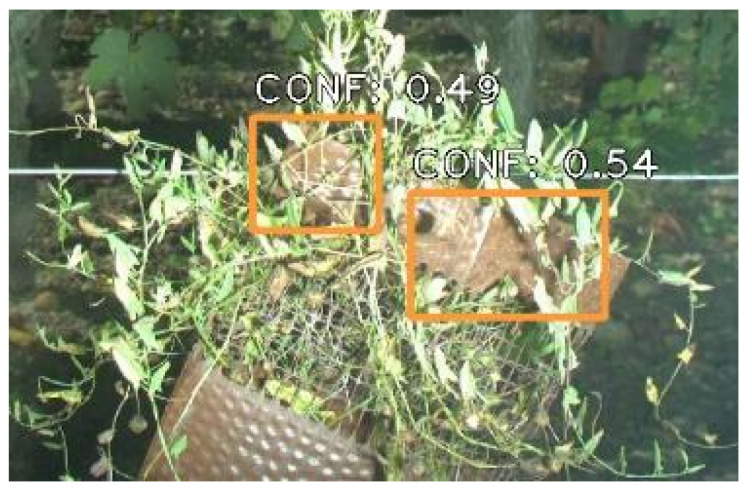
Extract of an image from the ADVANTAGE esca detection dataset showcasing false detections of dried leaves (orange boxes). Numbers indicate the algorithm confidence.

**Figure 10 sensors-23-00847-f010:**
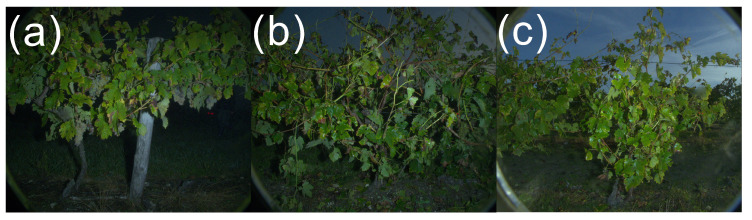
Example of the illumination evolution effect on the pictures taken over the course of the morning (from (**a**–**c**)) using a flash device with the same settings. Images extracted from the flavescence dorée detection ProspectFD dataset.

**Figure 11 sensors-23-00847-f011:**
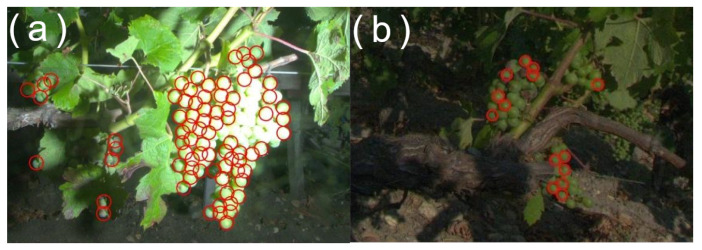
Example of results from the EARN dataset for the detection of berries (red circles). (**a**) Example of overexposed picture with undetected berries. (**b**) Example of a picture with no flash, showcasing undetected berries.

**Figure 12 sensors-23-00847-f012:**
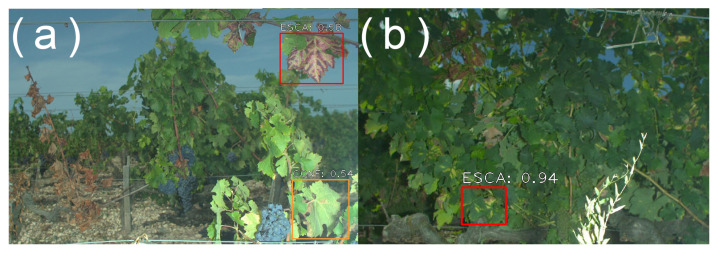
Example of detection results from the ADVANTAGE esca disease dataset. Red boxes: esca detections. Orange boxes: other symptoms and deficiencies detections. (**a**) Example of good behavior: background-row dried leaves appropriately not detected. (**b**) Example of bad behavior: background-row esca leaf detected. Figures in both images indicate the algorithm’s confidence.

**Figure 13 sensors-23-00847-f013:**
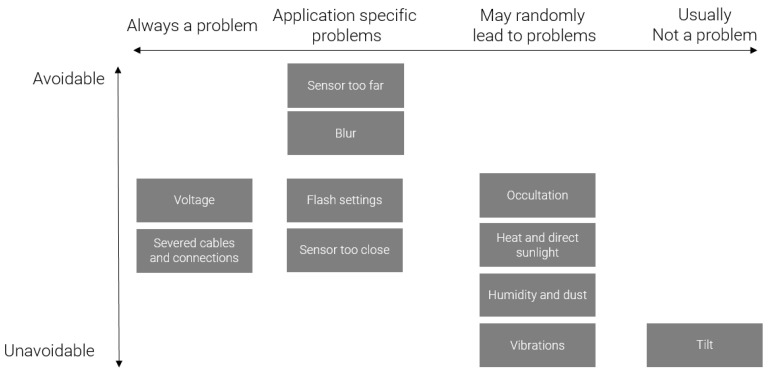
Synthesis of the acquisition parameters. The horizontal axis describes how crucial a parameter is to the acquisition. The vertical axis describes the difficulty in controlling this parameter.

**Table 2 sensors-23-00847-t002:** Methodological detail of the algorithms used to process the images acquired by the experimental setup on different projects.

Task and Reference	Methodology and End Result	Approximate Number of Images/Annotated Images	Visual Result Example
Berry detection [27]	Methodology: Detection of gradients on round objects illuminated by the flash End results Bounding circles	∼250,000/150 (∼10,000 berries)	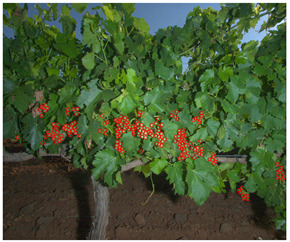
Esca symptoms detection [28]	Methodology: SIFT feature extraction combined with bag-of-word modeling; convolutional neural network backbone integrated to the MobileNet detection network End results: Bounding boxes	∼1800/∼1200	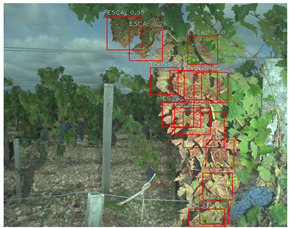
Downy Mildew symptoms detection [29]	Methodology: Color structure tensor modeling combined with hysteresis segmentation End results: Segmentation (manually circled in the result example)	∼10,000/∼400	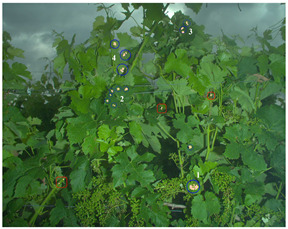
Flavescence Dorée detection [30]	Methodology: Convolutional neural network associated with a random forest algorithm for decision at the plant scale End results: Bounding boxes/decision at the plant scale	∼43,000/∼1400 (image-scale label) + 1000 (leaf-scale bounding boxes)	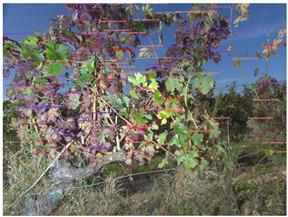

**Table 3 sensors-23-00847-t003:** Description of the vectors on which the acquisition system may be mounted.

Vector	Yield/Speed	Availability
Harvesting tractor	High/constant speed	Harvest and sometimes tillage works; skilled trained driver
High-leg tractor	High/constant speed	During crop control/tillage works; skilled trained driver
Within-row tractor	High/constant speed	Trained driver
Remote-controlled robot	Medium/variable speed	Anytime access is possible; trained user
“Pedestrian” System	Low/variable speed, user fatigue	Anytime access is possible; anyone

## Data Availability

Data availability for each individual project: EARN (yield estimation): available through the article author (barna.keresztes@ims-bordeaux.fr); ADVANTAGE (esca detection): available through the corresponding author (florian.rancon@agro-bordeaux.fr); ProxiMaVi (downy mildew detection): available online through a published data article [39]; ProspectFD: available through the article author (malo.tardif@u-bordeaux.fr).

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
