# Peer review of "Designing a Proximal Sensing Camera Acquisition System for Vineyard Applications: Results and Feedback on 8 Years of Experiments"

_sensors, 2023, doi:10.3390/s23020847_

Round 1

Reviewer 1 Report

The paper aims to design a mobile camera for vineyards and harsh environments. The authors are sharing their experience with the scientific community; however, the topic is still missing many essential elements which were not presented in the current manuscript. The main comments are given below:

1-      The paper is not focused on one topic, which makes it difficult to follow. It would contribute more if other research works from others were also compared, not only the author's own view.

2-      This work is based on the use of the mobile camera from the authors' previous work in [19], [20], [21], [24], [27], and [28]. Reading the current manuscript without going back to such references makes it difficult to follow.

3-      The pictures from previous work discussed specific problems such as fruit/symptoms detection, making the reader go back to understand the scope of using the camera and the meaning of the numbers on such figures.

4-      It was very difficult to follow the Figures in the manuscript. All Figures (Fig. 2, Fig. 3, Fig. 4, Fig. 5, Fig. 7, Fig. 8, Fig. 10, Fig. 11, and Fig. 12) need to be numbered, for example, (Fig.2a and Fig.2b), (Fig.3a, Fig.3b, Fig.3c, Fig,3d). All the changes should be reflected in the manuscript.

5-      It would be a significant benefit for the scientific community if the authors shared the image data, which in my opinion, will be the main contribution of this work. Such open access dataset will enable other work to be compared and will be more valuable.

6-      The problem of image localization was not presented nor discussed in the current manuscript.

7-      Although the title is “design a mobile camera”, only figures 1 and 2 show a simplified schematic without much detail for the circuit diagrams. Other details for connections/configurations that other researchers could benefit from are missing.

Other comments:

-          About the language, extensive editing of the English language and style required is required.

-          Many typo errors, such as:

*Page 7, line 221, in in….

* Lines 133-145, this section needs to be rewritten as one paragraph.

* change “e.g.” to “e.g.,” in line 86, 256,307, 327, 384, 397,402, etc.

* Line 456, …., it is

* Reference 21 is the author's work but no more information was given for the conference/journal name and date.

Reviewer 2 Report

The manuscript deals with an analysis, description and discussion of common problems related to the use of images for crop sensing. Authors focus on grapevine during the vegetative cycle, color (RGB) imaging, proximal sensing and mobile acquisition and mention, including in the title, that it represents eight years of experiences.

The contribution is interesting and useful for those that are entering in this area of study, but quite obvious for those experienced ones. It is unavoidable to mention that it is not exactly a scientific paper, but a narrative set of experiences that would fit well in a more educational or extension kind of publication. There is no exactly Results and the Discussion does not provide citations to support the authors opinion, as the results are difuse.

Some more specifics:

Ln 87 – “However the acquisition step is also known to be very difficult, agriculture being conducted most of the time in non-controlled environments.” – something is missing here.

Ln 117 b to 126 – After “:”, several paragraphs (with more than one sentences) separated by “;”. Not a classical text structure. ( same on Ln 263 to 274).

From Ln 181, GNSS need to be more technically specified; any augmentation system?

Ln 221 - in in

Round 2

Reviewer 1 Report

Thank you for addressing my comments and concerns that I have provided in round 1. Please re-check the English language, write as one paragraph (lines 179-189), and correct ( Figure 11a presents an example...) in line 496.

Reviewer 2 Report

On line 520 - Fosr instance...